# Carotenoid Content in Human Colostrum is Associated to Preterm/Full-Term Birth Condition

**DOI:** 10.3390/nu10111654

**Published:** 2018-11-03

**Authors:** Ana A. O. Xavier, Elena Díaz-Salido, Isabel Arenilla-Vélez, Josefa Aguayo-Maldonado, Juan Garrido-Fernández, Javier Fontecha, Alicia Sánchez-García, Antonio Pérez-Gálvez

**Affiliations:** 1Department of Food Phytochemistry, Instituto de la Grasa (CSIC), Campus Universitario, Building 46, 41013 Sevilla, Spain; anaaugustax@gmail.com (A.A.O.X.); jgarrido@ig.csic.es (J.G.-F.); 2Unidad de Neonatología, Hospital Virgen del Rocío, 41013 Sevilla, Spain; elenam.diaz.sspa@juntadeandalucia.es (E.D.-S.); Isaareni@yahoo.es (I.A.-V.); pepaaguayo@gmail.com (J.A.-M.); 3Institute of Food Science Research (CSIC-UAM), 28049 Madrid, Spain; j.fontecha@csic.es; 4Laboratorio de Espectrometría de Masas, Instituto de la Grasa (CSIC), 41013 Sevilla, Spain; asangar@ig.csic.es

**Keywords:** breastfeeding, new-born, full-term mothers, preterm mothers, xanthophylls, carotenes, lutein, colostrum, mature milk

## Abstract

Factors such as lactation stage and premature and small-for-gestational conditions could lead to great inter-individual variability in the carotenoid content of human milk. The aim was to analyze the carotenoid content in colostrum and mature milk of preterm (PT) and full-term (FT) mothers to establish whether they are significantly different and, if so, the stage of lactation when the differences are established. Samples of blood, colostrum, and mature milk were collected from Spanish donating mothers who gave birth to PT or FT infants. Carotenoids from serum and milk samples were analyzed by HPLC-atmospheric pressure chemical ionization (APCI)-MS. Quantitatively, colostrum from PT mothers presented lower total carotenoid content when compared to that from FT mothers. The only exception was lutein, where levels were not different. The transition from colostrum to mature milk makes observed differences in the carotenoid content disappear, since there were no variances between PT and FT groups for both individual and total carotenoid content. The premature birth condition affects the quantitative carotenoid composition of the colostrum but has no effect on the lutein content. This fact could be related to the significant role of this xanthophyll in the development of infant retina and feasibly to cognitive function.

## 1. Introduction

Breastfeeding provides health advantages for the immediate growth and development of the new-born, benefits that will extend into adulthood [1], and decreases the risk of child morbidity and mortality [2,3]. International policy agencies recommend exclusive breastfeeding for six months, followed by continued breastfeeding with appropriate complementary foods for up to two years or beyond [4]. Breast milk contains the macro and micronutrients necessary to fulfil nutritional requirements of the infant, as well as immune and nonimmune properties and growth factors [5,6]. Many of these constituents in the milk are derived from maternal blood, which is enriched through diet. This is the case of carotenoid pigments that are exclusively synthesized de novo by photosynthetic organisms, bacteria, yeasts, and some fungi, and incorporated into human tissues and blood through food sources, fruits, vegetables, algae, eggs, and fish [7]. The human milk carotenoids with provitamin A activity contribute to the vitamin A needs of the infant, a key issue in those developing countries where the dietary supply of vitamin A sources is often limited [8]. Independently of the provitamin A activity, carotenoids exert important functions in immunity, participate in the antioxidant defense system, and are related to a reduced risk of developing chronic diseases [9,10]. The specific accumulation of lutein and zeaxanthin in the fovea of the retina is of remarkable interest, given that this tissue continues to develop during infancy [11]. These carotenoids, known as macular pigments, prevent retinal damage through their blue light filtration and antioxidant properties [12,13,14], and contribute to the proper structural development of the retina [15]. Furthermore, it has been shown that enriching the macular pigment density by lutein supplementation positively impacts on contrast sensitivity and glare disability [16]. Recent research correlates macular pigment content with cognitive function through different evidence, as it has been shown that the increase in visual performance reached with lutein supplementation is associated with better cognition [17,18], in a similar fashion as the levels of macular lutein and zeaxanthin are related to brain lutein and zeaxanthin content [19]. Also, it would be important to point out that these levels were significantly related to pre-mortem measures of cognitive function [20,21].

These statements may apply in the prenatal and postnatal developing infant. Vishwanathan et al. [22] have shown that lutein is the predominant carotenoid in the infant brain, with much lower levels of zeaxanthin, β-cryptoxanthin, and β-carotene. This accumulation is the starting point to support that carotenoids contribute to early neurodevelopment [23,24]. It could be assumed that the accumulation of carotenoids in the infant brain behave in a similar fashion to the macular pigments in the infant retina, where the carotenoids tend to accumulate in the final months of gestation [25,26]. Indeed, the content of macular pigments is variable in the developing retina [27] and such variability would extend to their brain content if both quantities are correlated as noted above. 

The macular and brain carotenoid levels after delivery are a consequence of the infant intake of carotenoids from colostrum and mature milk or infant formula, because these are the only dietary sources containing carotenoids that the new-born receives [23]. Also, it is well established that there is a high-amount of inter-individual and intra-individual variability in milk carotenoid content, which it is associated to maternal dietary habits [28,29], lactation stage [30,31,32,33], and health status of the mother, including alcohol intake and smoking [34,35]. With this knowledge, we are beginning to understand the factors contributing to the efficiency of the carotenoid accumulation in infant tissues during the initial months of age. In addition, some inputs are now identified regarding the similarities and differences associated with the carotenoid transfer from maternal blood with the transport of bulk lipids during lactation. Even the existence of transport mechanism of xanthophylls [36] and an active acylation pathway in the mammary glands [37] have been observed. Other features influencing the carotenoid status of the infant at birth, which are now under analysis, are the premature and small-for-gestational conditions. The highest utero accretion of energy and nutrients takes place during the last weeks of pregnancy and prematurity impairs the accumulation of carotenoids into the tissues of the child as shown by Vishwanathan et al. [22], who reported lower carotenoid content in brain of preterm (PT) infants compared to those born at full-term (FT) condition.

The aim of this work was to analyze the carotenoid content in the colostrum and mature milk of preterm and full-term mothers to determine whether they are significantly different and, if so, the stage of lactation when the differences are established and whether they are transitory or not. Prematurity is not a condition only affecting the developmental and nutritional stages of the infant, but also the maturity of the mammary gland and its ability to secrete milk with the appropriate composition for the situation of the new-born. An incomplete gestational period affects the deposition of nutrients in the mammary gland—as it has been described for the accumulation of fatty acids [38]—and the deficiencies may happen at birth or appear during the lactation transition to mature milk [39]. Thus, the incomplete maturity of the mammary gland could be a significant factor that may affect the potential of the breastfeeding to compensate for the nutritional deficits of the PT child. Neither the carotenoid content of the colostrum nor the dynamics of carotenoid content changes occurring in mothers who gave birth prematurely and the comparison with mothers whose children were born at full-term, have been analyzed so far. 

## 2. Materials and Methods

### 2.1. Subjects

The study population comprised 144 healthy women, which were recruited within a 12 month period between 2015 and 2016 and classified into two groups: mothers (*n* = 72) who gave birth to full-term neonates (37–40 weeks) and mothers (*n* = 72) who gave birth to preterm neonates (28–35 weeks). Eligible participants in this study were non-smoking mothers with no chronic diseases. Mothers following any special diets, vegetarians, or those taking supplements were not included. Exclusion criteria applied were pathologies and/or infections during the gestation, developmental anomalies in the fetus, or death of the child. They provided informed consent to participate in the research program and none of the initially enrolled mothers dropped out of the study.

Our previous study [37] indicated that 70 volunteers per group would be necessary to detect significant differences of the individual carotenoid profile between preterm and full-term delivery conditions and between lactation stages at a significance level of *p* = 0.05.

### 2.2. Measures

#### 2.2.1. Milk and Blood Sample Collection

The samples were collected at the Unidad de Neonatología of the Hospital Universitario Virgen del Rocío (Seville, Spain). Preterm and full-term mothers donated colostrum at 3–5 days postpartum and the same mothers donated mature milk at 30 days postpartum. Milk samples were obtained by collection of the total milk volume of one breast during one milk expression session into a polypropylene bottle. Fasting blood samples (10 mL) were collected at the same time of milk samples. Blood was centrifuged after clotting for 10 min at 2000× *g* at 4 °C to obtain serum. The samples were transported directly to the laboratory and stored at −80 °C until analysis.

#### 2.2.2. Extraction of the Carotenoid Fraction

The experimental conditions previously described by Ríos et al. [37] were used for extraction of carotenoids from human milk samples. Human milk (3 mL) was mixed with 3 mL of KOH:methanol (20% *w*/*v*), and the mixture was incubated for 1 h. After hydrolysis, 6 mL of methanol were added, and the mixture was vortex-mixed for 2 min and cooled at −20 °C for 20 min. Subsequently, the cooled mixture was centrifuged at 10,000× *g* and 4 °C for 5 min, discarding the upper layer. Diethyl ether (5 mL) and hexane (2 mL) were added to the pellet and vortex-mixed for 2 min. Then, 5 mL of NaCl 10% (*w*/*v*) was added, and the sample was vortex-mixed again for 2 min. After centrifugation (10,000× *g* at 4 °C for 5 min), the organic layer was washed with water until neutral pH was reached. The organic extract was evaporated to dryness in a rotatory evaporator at 25 °C, and the residue was dissolved in 1 mL of methanol:methyl tert-butyl ether (8:2). This solution was filtered through a 0.22-µm filter and stored at −20 °C until analysis. Carotenoid extraction from serum samples was carried out according to the method described by Pérez-Gálvez et al. [40]. Serum (0.1 mL) was extracted with 6 mL hexane:dichloromethane (5:1). The mixture was vortex-mixed for 1 min and centrifuged at 2000× *g* at 20 °C for 10 min. A portion of the upper phase (5 mL) was withdrawn and the solvent evaporated under N_2_. The extract was dissolved in 100 µL of acetone and stored at −20 °C until analysis by HPLC-MS, which was performed within 1 week.

#### 2.2.3. Identification and Quantification of Carotenoids in the Carotenoid Extracts 

The carotenoids from human milk and serum samples were separated using a Dionex Ultimate 3000RS U-HPLC (Thermo Fisher Scientific, Waltham, MA, USA) using the method developed by Breithaupt et al. [41] with slight modifications [37]. High-resolution mass spectrometry measurements were completed on the basis of mass accuracy and in combination with the isotopic pattern in the SigmaFit algorithm [37]. The characteristics of experimental mass spectrum and the MS^2^ data were compared with the data available in the literature for carotenoid identification in human milk and serum samples [42,43,44,45,46,47,48]. For carotenoid quantification, stock solutions of β-carotene, β-cryptoxanthin, lutein, and lycopene were prepared at a concentration of 25 mg/L. Once the exact concentration was determined (2% maximum total error), working stock solutions for external calibration curves were prepared at 5 concentration levels ranging from 0.15 to 10.0 mg/L. The content of xanthophylls and carotenes was determined in the unsaponified extract. Zeaxanthin was quantified with the calibration curve of lutein, while xanthophyll esters were quantified as their corresponding free xanthophyll [47,48].

#### 2.2.4. Determination of the Lipid Content

The lipid content of human milk samples was determined according to the solvent extraction procedure and then by gravimetry [49].

### 2.3. Ethics Approval

All subjects gave their informed consent for inclusion before they participated in the study, which was conducted in accordance with the Declaration of Helsinki. The study protocol was approved by the Ethics Committee of the Hospital Universitario Virgen del Rocío and the Bioethics Subcommittee of the Spanish National Research Council (AGL2013-42757R).

### 2.4. Statistical Analysis

Data are reported as the median, including 25th and 75th percentiles. Due to the non-normality of content of individual carotenoids (Kolmogorov–Smirnov test), the data were analyzed using a non-parametric statistical procedure in the SPSS software (IBM^®^ SPSS^®^ Statistics version 24, IBM, New York, NY, USA). The Mann–Whitney test was applied to analyze the differences in carotenoid content of colostrum and mature milk of PT and FT mothers, and to compare the carotenoid content of colostrum and mature milk within each group of lactating mothers. The same test was applied for carotenoid content in serum samples of both groups and at both collection stages (colostrum and mature milk). The significance was set at *p* < 0.05.

## 3. Results

The carotenoid composition of the colostrum and mature milk samples from PT and FT mothers is shown in Table 1. Colostrum samples from both groups were qualitatively equal regarding the carotenoid profile, but significant differences were observed when the groups were compared quantitatively. Specifically, the colostrum of FT mothers showed higher carotenoid content, both for individual carotenoids and for the total amount (median of total carotenoid content of 4961.1 nM in the colostrum of full-term mothers vs. 2641.1 nM in the colostrum from preterm mothers, *p* < 0.05), except for the lutein content that was not different between groups (486.31 nM vs. 432.83 nM, *p* = 0.238). Figure 1 shows the percentage distribution of the carotenoids in colostrum and mature milk of both groups of mothers. Xanthophylls, which are carotenoids containing oxygenated functions in their structure, and carotenes, which are strictly hydrocarbons, were similarly accumulated in colostrum samples of PT and FT mothers, as the ratio between both groups of pigments was close to 1 (0.97 and 0.90 for the preterm and the full-term group, respectively). The ranking of individual carotenoids in the colostrum samples was recurrent for both groups, with zeaxanthin presenting the lowest content value followed by the pool of xanthophyll esters (sum of lutein, zeaxanthin, and β-cryptoxanthin esters), the free xanthophylls lutein and β-cryptoxanthin, and finally by the carotenes α- and β-carotene and lycopene. The total fat content of colostrum was not related with the PT or FT condition, with values in the range of 35–45 mg/mL showing no significant differences. Regarding the carotenoid content of the mature milk, both individual and total amounts of carotenoids from PT mothers did not differ from that of FT mothers, either qualitatively or quantitatively. Thus, the total carotenoid content was not statistically different (median 768.0 nM vs. 693.54 nM, *p* = 0.278). The ratio of xanthophylls/carotenes in the mature milk samples was higher than 1 (1.61 and 1.70 for the FT and the PT groups, respectively), unlike the distribution pattern of carotenoids found in the colostrum samples. Zeaxanthin was the minor carotenoid present in the mature milk samples, likewise in the colostrum samples, but the xanthophyll esters were undetectable in both groups. Then, α- and β-carotene and lycopene followed, and finally β-cryptoxanthin and lutein. Again, the total milk fat content was similar in both groups, similar to the values reached in colostrum samples (30–40 mg/mL) and unrelated with the PT or FT condition, with no significant differences between groups.

Evolution of the carotenoid content from colostrum to mature milk followed a similar sharp decline but it was lower for the carotenoid content in mature milk of PT mothers (74% of the total carotenoid content) than for the FT mothers (85% of the total carotenoid content). Significant differences were observed when individual and total carotenoid contents of colostrum and mature milk samples were compared within each group of lactating mothers, except for zeaxanthin content of the group of PT mothers.

No significant differences were observed in both groups both for individual and total carotenoid content, as shown in Table 2 and Figure 1, in the serum from lactating mothers, either at the initial lactation stage (3–5 days after delivery) or once the milk reached the mature state (30 days after delivery). Within each group and during the progress of lactation, individual and total carotenoid content did not differ also, as shown in Table 2 and Figure 1. Xanthophylls and carotenes were similarly distributed in both groups of lactating mothers and this ratio was the same at both stages of lactation. Zeaxanthin was the minor carotenoid found in the serum, and the close similarity in the data of the rest of carotenoids makes it tough to establish a clear ranking.

## 4. Discussion

Individual and total content of the major carotenoids in human milk samples have been determined in previous studies [28,29,32,33,36], and our data are in the same order of magnitude and follow similar changes and trends regarding the transition from colostrum to mature milk and the distribution pattern of individual carotenoids. We also noted the high variability of data observed before. Giuliano et al. [28] showed that although the carotenoid content might fluctuate daily and even during feeding, the large variability in carotenoid content is due to inter-individual differences in dietary intake of food sources of carotenoids, the carotenoid bioavailability, and the bioconversion of pro-vitamin A carotenoids to retinoid. Indeed, the bioconversion to retinoid in addition to the nutritional status of the mother may explain the extreme inter-individual differences in the case of α- and β-carotene. 

Studies focused on carotenoid content in human colostrum are less frequent in the literature [30,36,37,50], and the characterization of the full carotenoid profile in some of them is incomplete, in comparison with the extensive surveys done with mature human milk samples. Colostrum is a secretion of the mammary gland only available during the few days immediately after delivery, and its composition does not represent the stabilized nutrient contents of mature milk. These are reasonable causes to explain the less intensive attention paid to carotenoids in human colostrum. Most longitudinal studies do not include colostrum collection in the sampling, while some others do. In the latter cases, data should be considered carefully as the carotenoid content in colostrum is significantly higher than in mature milk and even the distribution of carotenoids (xanthophylls, carotenes, and presence of xanthophyll esters) may be different as well [37]. It is well established that carotenoid levels decrease significantly from the colostrum to the mature milk samples as observed in preceding longitudinal and comparative studies [32]. We found a drop of carotenoid levels similar (4-fold for PT mothers and 6-fold for FT mothers) to those reported by Sommerburg et al. [51], Macias and Schweigert [52], and Schweigert et al. [36].

The total carotenoid content observed for mature milk in this study are in the highest range of data reported for North American mothers [28], close to the mean values reported for Irish mothers [53], and higher than those found in a study on milk of German mothers [36]. Finally, the data presented in this study are higher when compared with the largest multinational study of breast milk carotenoids published by Canfield et al. [29]. The main reasons for the differences and similarities are dietary habits including individual preferences, seasonal/regional changes in fruit and vegetable availability, and cooking methods. Qualitatively, the carotenoids were distributed in colostrum samples in a different fashion in comparison to mature milk, with an almost equal presence of carotenes and xanthophylls, while this equilibrium is significantly displaced to the preferential accumulation of xanthophylls during the progress of lactation in agreement with previous reports [32,36]. Mature milk showed a distribution pattern with the prevalence of the xanthophylls lutein and β-cryptoxanthin, followed by the apolar carotenes lycopene and α+β-carotene. Zeaxanthin was a minor carotenoid. This displacement is not correlated with the carotenoid distribution in plasma samples, which was the same at both collection stages, with similar distribution of carotenes and xanthophylls, as shown in Figure 1. This constancy of carotenoid distribution and quantitative content in plasma could indicate that the volunteers had similar food sources of carotenoids throughout the study and that they did not introduce significant changes in their dietary habits after delivery. Circulating carotenoid content is positively associated with the daily intake of fruits and vegetables. It has been shown that qualitative carotenoid composition of serum correlates with short-term carotenoid intake [54,55] and particularly, β-cryptoxanthin and lutein are robust biomarkers of fruit and vegetable consumption [56]. Therefore, the shift of carotenoid distribution towards a higher xanthophyll presence in mature milk would not likely be due to a change in dietary habits. It seems that during the transition of the carotenoid content from colostrum to mature milk, xanthophylls are preferentially accumulated in the mammary epithelium. Transport mechanisms operative at the intestinal epithelium that are involved in the uptake of carotenoids from the lumen, may be active at the mammary epithelium. In fact, it has been demonstrated that SR-BI, CD36, and NPC1L1 facilitate the uptake of carotenoids in the small intestine [57,58,59], and these proteins are also expressed in the mammary glands [60,61]. Even the expression of some of these proteins has been shown to increase in the mammary tissue of lactating animals. This is the case of CD36 in lactating cows [62] which increases its expression levels after delivery, reaching a maximum at 6 weeks postpartum. Whether such an increase in the expression levels of any of these proteins take place coordinately after delivery in the human mammary gland, and whether they are specific for some protein kind remains to be elucidated, though this would help to explain differences in the xanthophyll to carotene proportion in colostrum and mature milk.

Regarding the influence of the gestational age on the carotenoid content, we observed that significant differences appeared only in colostrum samples. The group of PT mothers produced colostrum with lower carotenoid levels, except for lutein that reached a similar content to the one observed in the group of FT mothers. The carotenoid distribution in colostrum was similar in both groups (equal xanthophyll:carotene ratio), so that the differences were only quantitative. Jewell et al. [53] reported differences in lutein and zeaxanthin content in colostrum between PT and FT mothers, but the statistical power was limited in their study and differences were not significant. The differences observed here should not be attributed to diverse dietary habits of the mothers of the PT group, since the plasma carotenoid contents were not different from those observed in the mothers of the FT group. Several changes take place in the mammary glands during pregnancy to prepare for lactation, including gland maturation and alveologenesis [63], with the start of several signaling pathways that develop the structural and functional changes required to generate and deliver milk. The maternal metabolism is remodeled to flow nutrients to the placenta and the mammary glands for support fetal and infant growth. In this scenario, the premature delivery breaks the regular process of utero accretion and transfer of nutrients to the developing fetus, as well as the accumulation of nutrients and phytochemicals in the mammary glands from the mother stores. It has been shown that carotenoid content in PT infants’ tissues is abnormally low or undetectable [26]. Our data show the differences in the carotenoid contents are also extended to the colostrum secretion in the case of premature delivery, with a trend to lower carotenoid content. However, it does not affect the lutein content in colostrum of PT mothers, which reached the same value as in the group of FT mothers. This fact could be related with the significance of lutein to the health of the new born. Lutein content is associated with proper condition of the infant retina [64] and its accumulation in the brain points to a role in cognitive function [21]. Such impact at the initial stage of life implies a mechanism assuring the lutein supply from human milk. Thus, it would not be impaired by factors like the premature condition. Again, the presence of a facilitated transport mechanism, principally active for lutein accumulation, is a reasonable hypothesis to explain this result. As it has been noted above, the measurement of the levels of those proteins involved in carotenoid transport in mammary glands would provide evidence to support this hypothesis. The subsequent regularization process of the carotenoid content from colostrum to mature milk is not affected by the premature condition since the mature milk of both groups of mothers showed carotenoid levels without significant differences both qualitatively and quantitatively.

To our knowledge, this is the first study to examine and compare the qualitative and quantitative carotenoid profile in colostrum and mature milk samples from lactating mothers that gave birth either prematurely or at full-term. The homogeneous sample, regarding the dietary habits, controlled one of the main factors that determine carotenoid levels in human tissues, including colostrum and milk samples. The qualitative findings replicated those reported in previous studies, but through the inclusion of an adequate number of volunteers, we had statistical power to find the differences in carotenoid content of the colostrum samples. Thus, our data give support to the hypothesis of the targeted accumulation of lutein and that this process is not limited by the premature condition at birth. Indeed, this condition reduced the amount of the rest of carotenoids in colostrum but the subsequent transition stage to mature milk made differences disappear. However, we are not able to predict whether our findings will reproduce in other populations with different dietary habits.

## 5. Conclusions

In our study, we have determined the qualitative and quantitative features of the carotenoid composition of colostrum and mature milk of mothers who gave birth to babies born between 28 and 35 weeks of gestation and to infants at full-term. Our results have shown that the qualitative profile does not vary between groups, but there is a shift from a balanced composition in carotenes and xanthophylls in colostrum to a higher presence of the latter in mature milk. Quantitatively, we found significant differences between the groups in the colostrum samples, with lower content in the preterm group, except for lutein, which behaves in a different fashion. The transition from colostrum to mature milk results in no difference in the carotenoid content between the two groups. These results lead to the conclusion that the premature condition affects the quantitative carotenoid composition of the colostrum, but that condition does not impair the lutein content, which is related to the significant role of the xanthophyll in development of the infant retina and brain. Our study points to the need of research regarding the presence of protein-type transporters in mammary epithelium as one key issue for the existing dynamics on carotenoid content in human milk.

## Figures and Tables

**Figure 1 nutrients-10-01654-f001:**
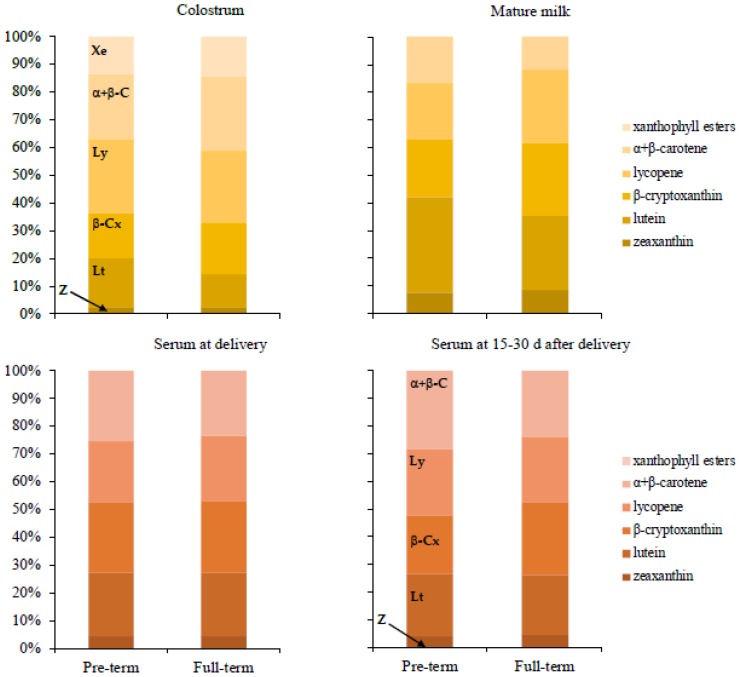
Percentage distribution of the individual carotenoids and xanthophyll esters in colostrum and mature milk samples, and in serum collected at delivery or at 15–30 days after delivery, from preterm and full-term mothers. Xe: xanthophyll esters; α+β-C: α+β-carotene; Ly: lycopene; β-Cx: β-cryptoxanthin; Lt: lutein; Z: zeaxanthin.

**Table 1 nutrients-10-01654-t001:** Carotenoid content in colostrum and mature milk of preterm (PT, *n* = 70) and full-term (FT, *n* = 70) mothers. Data are expressed in nM concentration.

	PT Mothers ^1^			FT Mothers ^1^		
	25th Percentile	Median	75th Percentile	25th Percentile	Median	75th Percentile
Colostrum ^2^						
Zeaxanthin (X) ^3^	35.9	63.2	112.6	73.7	106.4	141.4
Lutein (X)	231.6	432.8 ^a^	667.9	322.9	486.3 ^a^	745.8
β-Cryptoxanthin (X)	231.4	406.7	853.0	429.6	754.6	1486
Lycopene (C) ^4^	388.1	669.9	931.6	483.0	1065	1846
α+β-Carotene (C)	331.0	594.6	1364	602.3	1103	2238
Xanthophyll esters (X)	144.5	334.6 ^b^	733.4	251.5	598.7 ^b^	1211
Total	1810	2641	4498	2854	4961	7209
Mature milk ^5^						
Zeaxanthin	36.7	46.6	65.9	38.4	59.9	91.9
Lutein	170.0	217.3	283.0	150.3	195.9	270.8
β-Cryptoxanthin	53.0	135.1	224.6	87.4	190.6	353.9
Lycopene	89.4	125.8	173.6	118.8	192.7	221.2
α+β-Carotene	50.0	106.4	202.2	59.4	84.8	315.0
Xanthophyll esters	0	0	0	0	0	0
Total	508.7	693.5	934.0	512.3	768.0	1174

^1^: Data are significantly different when colostrum and mature milk values are compared within each group of lactating mothers, except for zeaxanthin of the preterm group. ^2^: Data of colostrum samples were significantly different between both groups of lactating mothers except for values marked with superscript letters (*p* < 0.05). ^3^: X means xanthophyll. ^4^: C means carotene. ^5^: No significant differences were observed for data of both groups of lactating mothers.

**Table 2 nutrients-10-01654-t002:** Carotenoid content in serum of preterm (PT, *n* = 70) and full-term (FT, *n* = 70) mothers at two lactation stages. Data are expressed in nM concentration.

	PT Mothers			FT Mothers		
	25th Percentile	Median	75th Percentile	25th percentile	Median	75th Percentile
Serum samples collected at 3–5 day postpartum ^1^						
Zeaxanthin	108.2	142.9	171.8	94.4	122.1	152.4
Lutein	658.3	744.8	911.9	552.4	656.0	834.7
β-Cryptoxanthin	419.8	814.1	964.7	617.8	737.3	1084
Lycopene	481.7	712.9	980.2	450.3	673.5	887.3
α+β-Carotene	519.6	816.8	1089	494.6	671.2	1547
Xanthophyll esters	0	0	0	0	0	0
Total	2984	3288	3751	2869	3222	3811
Serum samples collected at 15–30 day postpartum ^1^						
Zeaxanthin	105.3	136.1	187.4	96.0	135.5	158.6
Lutein	622.6	768.4	1010	579.0	673.9	809.1
β-Cryptoxanthin	329.9	729.3	1152	669.2	822.4	925.2
Lycopene	555.3	824.0	1007	473.9	730.4	905.7
α+β-Carotene	583.3	967.4	1594	549.2	743.8	1087
Xanthophyll esters	0	0	0	0	0	0
Total	3279	3636	3898	2938	3295	3711

^1^: No significant differences were observed for data of both groups of lactating mothers or at each collection time within a group.

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
