# Peer review of "Carotenoid Content in Human Colostrum is Associated to Preterm/Full-Term Birth Condition"

_nutrients, 2018, doi:10.3390/nu10111654_

Round 1
Reviewer 1 Report
Its a well-written manuscript which could be improved with the suggested suggestion
Introduction provides enough background and is super long, yet the rationale for the study is weak. Clearly, describe the gap in the literature and describe the need for this study.
Method- despite citing the reference for milk extraction, having a brief description is essential for understanding the method.
When were the blood samples taken? Blood was taken at the same time as milk? this information is unclear
Include some of the limitations of your study? Did you take maternal dietary information, if so when and what information was collected?
Check if have used the latest references
Author Response
Answer to reviewer 1.
We have reviewed the manuscript according to your suggestions that improve the clarity of presentation and discussion of the results. Regarding the introduction section, we have included the relevant references published in the subject aiming to provide the reader sufficient background. The rationale for the study was detailed in lines 81-94 of the original manuscript. We have changed the statements to clarify this issue in the revised version.
The reviewed manuscript includes a brief description of the method applied for extraction of carotenoids from human milk samples, as well as for the extraction of carotenoids from serum samples.
Blood samples were collected during acquisition of milk samples. This information has been included in the revised version of the manuscript.
Maternal dietary information was not obtained and indeed this is the limitation of our study. Therefore and following the suggestions of reviewer 2, the statements made regarding this issue have been modified.
Reviewer 2 Report
Authors determined, in this study, carotenoid content in human milk at 2 times of lactation and in mother’s serum at similar times. The situation of prematurity is compared to normal term. Six classes of compounds are studied and this makes the study rather complete. This is an observational study with 72 mothers in each arm. The design is very simple but data are complete (although this would have been interesting to know the carotenoid content in infant serum). I have no main concern except that:
- This is not a very original topic with many publications already published (authors listed them in the bibliography section);
- The same authors published very similar data last year in reference 37 (same study’s design: colostrum and mature milk, preterm and full term). Data in this paper are a little bit more complete (preterm and term milks are differentiated) but there is no great novelty in the design except the sampling of mothers’ plasma.
In detail:
Line 23: “We believe it is related to the significant role of this xanthophyll in the development of infant retina and feasibly to cognitive function.”
& Line 297: “This fact may be interpreted as lutein being of importance the health of the new born.”
These 2 writings reflect a very anthropomorphic thought. Please change the sentences.
Line 96: “The study population comprised 144 healthy women” How long does it take for the recruitment? When were they recruited (beginning and end)? These questions are related to the similarity between this paper and reference 37.
Line 111: “Milk samples were obtained by collection of the total milk volume of one breast during one milk expression session into a polypropylene bottle.” I am surprised that the ethical committee accepted such a protocol for preterm infants especially for colostrum that is generally kept for infant nutrition. What volume was sampled?
Line 104: “Our previous study (37) indicated that 70 volunteers per group would be necessary to detect differences between…”. How much difference? What compound?
Line 120: “2.2.3. Identification and quantification of carotenoids in the carotenoid extracts” What is the total error (in %) on the method used for every compound?
This paragraph “2 2 3” is very similar to that in the reference 37 from the same authors and should be really simplified.
Line 138: “For carotenoid quantification, stock solutions of β-carotene, β-cryptoxanthin, lutein, and lycopene were…” What’s about zeaxanthine?
Line 141: “…prepared at 5 concentration levels ranging from 0.15 to 10.0 mg/L.” These concentrations seem very high compared to the data presented in Table 1 although this is difficult to compare with the units used. Is the answer linear? A Figure presenting calibration curves (with the same units as in Tables) would be helpful to check the accuracy of the data presented.
Line 172: “(0.97 and 0.90 for the pre-term and the full-term group, respectively)”. Only experts in carotenes know to which class belongs every compound determined. Please differentiate in Table 1 between classes of “xanthophyll” or “carotene”.
Figure 1 is difficult to read because colors are very similar. Maybe mark each box with a letter (Z, Lt, Cx, Lp, C, Xe) corresponding to the first letter of each compound. In serum xanthophyll esters can be excluded from the legend.
Line 257: “This constancy of carotenoid distribution and quantitative contents in plasma means that the volunteers had similar food sources of carotenoids throughout the study and that they did not introduce significant changes in their dietary habits after delivery.” This is only a hypothesis that has not been tested since no Food Frequency Questionnaire was used. Please do not write that like a statement.
Line 263 (and 325): “Therefore, the shift of carotenoid distribution towards a higher xanthophyll presence in mature milk is not likely due to a change in dietary habits.” This evolution seems very similar for both carotenes and xanthophylls between colostrum and mature milk and a little bit stronger for carotenes than for xanthophylls. Although this is apparently true, did authors test the statistical significance of this evolution?
Author Response
Answer to reviewer 2
Significance of human milk at the early stage of life of the new-born is a critical issue, so that research efforts made to date and in the near future come to support the activities and recommendations in this subject of the health national and international bodies and policymakers. Nevertheless, this is the first study regarding the qualitative and quantitative carotenoid profile in colostrum and mature milk samples from lactating mothers that gave birth either prematurely of at full-term with enough sampling to statistically determine differences (or not) between carotenoid content in colostrum and mature milk samples.
The reviewer is right. We published a pilot study with colostrum and mature milk samples, but the subject of that paper was the identification of new molecular entities in the carotenoid profile of human colostrum, the xanthophyll esters, while the approach in this manuscript was to compare the carotenoid profile (both qualitatively and quantitatively) of preterm and fullterm mothers, a completely different issue not referred so far.
Statements of lines 23 and 297 have been changed according to your suggestion.
The study was performed within one year to obtain the required amount of samples for statistical significance of data. The volume of one breast (4-20 mL for colostrum or 50-150 mL for mature milk) was donated. Only those mothers that satisfied the nutritional needs of their infants donated colostrum/mature milk, so that the health of the new-borns was not compromised in any circumstance.
The significance was set at 0.05 level and the differences were observed for the individual carotenoid profile.
A 2% maximum total error was established for quantification, and the paragraph 2.2.3 has been summarised according to the suggestion of the reviewer. Zeaxanthin was quantified with the calibration curve of lutein. The concentration range is right. The amount of carotenoids per 50 μL of injected sample is within the 5-400 ng range, so that the 0.15 mg/L and the 10.0 mg/mL stock solutions cover that range. A Figure with the linear calibration curves would not add substantial information to the data presented in this manuscript. Otherwise, if the Editor considers it is necessary we could include this data as supplementary files.
Definition of carotenes and xanthophylls has been included in the text and in Table 1 to help the reader distinguishing between both families of carotenoids.
Figure 1 includes the boxes marked with letters to identify the individual carotenoids.
The reviewer is right. We did not include that differences between xanthophylls and carotenes is significant in the mature milk, while the colostrum contains almost equal distribution of both families of carotenoids. This issue has been stated in the results section, and it is in agreement with previous references.